# Characteristics and Risk Factors for Electric Scooter-Related Crashes and Injury Crashes among Scooter Riders: A Two-Phase Survey Study

**DOI:** 10.3390/ijerph191610129

**Published:** 2022-08-16

**Authors:** Disi Tian, Andrew D. Ryan, Curtis M. Craig, Kelsey Sievert, Nichole L. Morris

**Affiliations:** 1HumanFIRST Laboratory, Department of Mechanical Engineering, University of Minnesota, Minneapolis, MN 55455, USA; 2Midwest Center for Occupational Health and Safety Education and Research Center, Division of Environmental Health Sciences, School of Public Health, University of Minnesota, Minneapolis, MN 55455, USA

**Keywords:** electric scooter, e-scooter, infrastructure selection, electric scooter-related crashes and injury crashes, injury characteristics, risk factors

## Abstract

Electric scooters (or e-scooters) are among the most popular micromobility options that have experienced an enormous expansion in urban transportation systems across the world in recent years. Along with the increased usage of e-scooters, the increasing number of e-scooter-related injuries has also become an emerging global public health concern. However, little is known regarding the risk factors for e-scooter-related crashes and injury crashes. This study consisted of a two-phase survey questionnaire administered to a cohort of e-scooter riders (*n* = 210), which obtained exposure information on riders’ demographics, riding behaviors (including infrastructure selection), helmet use, and other crash-related factors. The risk ratios of riders’ self-reported involvement in an e-scooter-related crash (i.e., any crash versus no crash) and injury crash (i.e., injury crash versus non-injury crash) were estimated across exposure subcategories using the Negative Binomial regression approach. Males and frequent users of e-scooters were associated with an increased risk of e-scooter-related crashes of any type. For the e-scooter-related injury crashes, more frequently riding on bike lanes (i.e., greater than 25% of the time), either protected or unprotected, was identified as a protective factor. E-scooter-related injury crashes were more likely to occur among females, who reported riding on sidewalks and non-paved surfaces more frequently. The study may help inform public policy regarding e-scooter legislation and prioritize efforts to establish suitable road infrastructure for improved e-scooter riding safety.

## 1. Introduction

Electric “micromobility” services, typically characterized as adopting small-sized personal mobility devices such as bikes or scooters to travel short distances, have grown enormously across major cities over the past few years within a number of countries in North America, South America, Europe, and the Oceania region [1,2,3]. The electric scooter (i.e., e-scooter) has been recognized as one of the most popular urban transport micromobility options due to its affordability, convenience, and environmentally friendly features, and was particularly favored by many to help alleviate the “first-mile/last-mile” mobility issue [3,4,5,6,7]. Despite being powered by sustainable energy consumption, an e-scooter can sometimes move as fast as 25 km/h, allowing it to travel conveniently in various traffic conditions and interchangeably on different road infrastructures [6,8,9]. In 2019, Americans accounted for an estimated 88.5 million e-scooter trips, representing an almost 130% increase compared to the 38.5 million rides with e-scooters in 2018 [10]. Nevertheless, along with the global expansion of personal e-scooter ownership, and station-based and dockless e-scooter sharing systems, the increasing number of e-scooter-related injuries has also become an emerging public health concern among riders and other road users across the world [1,10,11].

Researchers from many countries have examined the injury trends and patterns associated with e-scooter falls or collisions [12,13,14,15,16,17]. To date, the majority of the literature has focused on utilizing hospitalization data, such as emergency department visits or admissions to a trauma center, in a retrospective or prospective case analysis [11]. Younger males, especially if they are full-time workers, have a high education level, or are less satisfied with the current transportation options, are consistently suggested to have a greater intention to use an e-scooter [3,5,18,19]. Similarly, compared to females, males also comprised a higher proportion of the patient population admitted and treated for e-scooter-related injuries [13,15,17]. Among all types of hospitalized motorized scooter injuries, the most common clinical diagnoses have been identified as fractures or dislocations (commonly occurring in the upper extremities) and head contusions (often in conjunction with a lack of helmet use among e-scooter riders) [12,14,20,21]. Although the severity level of an e-scooter-associated trauma can encompass a wide range, additional factors such as alcohol intoxication and substance use in the patient’s profile may potentially exacerbate the attributable health care costs [20,22,23]. However, less research has examined general safety aspects, particularly e-scooter-related crash and injury risks, among the general e-scooter user populations (i.e., non-patients), which may impact public health and indicate risk factors that are not apparent from a clinical sample.

Another major challenge faced by urban transportation planners and policymakers lies in the ambiguity and non-conformity of legislation on e-scooter riding or micromobility. In many U.S. states and jurisdictions, e-scooters have been essentially considered as bicycles (e.g., Ohio, Kentucky, and Hawaii). However, some U.S. cities, states, and other countries have enacted more stringent regulations regarding e-scooter use, including restrictions on the rider’s age, operation speed, and time, along with requirements on equipment, accessibility, parking, helmet use, etc. [8,24,25]. Although several existing laws, regulations, or local ordinances prohibit e-scooter riding on sidewalks or highways, there is still no census about which type of road infrastructure an e-scooter should be operated on. Because e-scooters have been primarily used as a means of “leisure or fun” riding or as a substitution for walking and other public transit [6,19,26,27,28], riders typically choose to ride on shorter (e.g., trips averaging approximately 1 mile) and less complicated routes [10,29]. Riders are shown to disproportionally favor bikeways for longer-distance trips and reportedly felt safer riding in the street when presented with a protected bike lane [10,25,29]. However, some riders have also been observed to frequently and, depending on jurisdiction, illegally ride on sidewalks or in the direction opposite to the traffic flow, potentially revealing a considerable discrepancy between e-scooter users’ actual riding behaviors and their knowledge of regulations [8,20,30]. Furthermore, little is known about the association between e-scooter users’ riding behaviors, particularly their infrastructure selection, and the risk for e-scooter-related crashes and relevant injuries.

### Study Aims

This two-phase survey study aimed to identify the characteristics of and ascertain risk factors for electric scooter-related crashes and injury crashes, among a cohort of e-scooter riders (*n* = 210). The primary focus was to analyze the relationship between riders’ infrastructure selection for riding an e-scooter and their self-reported involvement in e-scooter-related crashes and injury crashes. Relevant associated factors such as rider demographics, helmet use, and other crash related factors were also examined. The study findings could serve as a research basis for ascertaining potentially modifiable factors for mitigating relevant crash and injury risks among riders and promoting safer riding with e-scooters.

## 2. Materials and Methods

### 2.1. Study Sample

This study utilized a cross-sectional study design to survey riding behaviors of e-scooter users from different countries and their self-reported involvement in e-scooter-related crashes of any type and those that incur an injury. A sample of *n* = 210 participants remained in the study out of a two-phase recruitment process (total *n* = 255). Eligibility criteria included those aged 18 years and above, who reported having ever ridden on a standing e-scooter, with normal vision, and English language reading comprehension (no restrictions on country in which the participant lived). Data (17.6%) were excluded if a respondent was identified as a non-e-scooter rider, failed to complete the survey, or had missing responses for the crash outcomes (see Figure 1).

### 2.2. Data Collection

The data collection period ranged from 26 February 2021 to 2 September 2021 for Phase I recruitment and from 30 November 2021 to December 2, 2021 for Phase II recruitment. Participants were recruited through posted announcements seeking research volunteers on multiple social media platforms and technology discussion forums specific to e-scooters or micromobility, such as various e-scooter Facebook groups, Reddit webpages, and the research laboratory Twitter. Upon recruitment, participants consented to the study and were provided with a secured link to an online survey questionnaire via the Qualtrics software using the University of Minnesota design scheme (i.e., headings, background, color, etc.). The overall duration for participants to complete all survey questions ranged from 5 to 20 min. No monetary compensation was associated with the study participation.

The survey questionnaire was developed based on a literature review of available e-scooter-related survey research [19,21,31], and investigations into the crash and injury characteristics involving e-scooters [12,13,14,15,16,17,32,33,34]. Participants were asked to respond to 116 single or multiple-choice questions regarding e-scooter riders’ demographic characteristics, self-reported riding behaviors, helmet use, and involvement in e-scooter-related crashes and injury crashes. The section below presents detailed definitions of variables, as measured in the survey questionnaire.

### 2.3. Definitions of Variables

The study outcomes consisted of two binary variables identifying riders’ self-reported involvement in e-scooter-related crashes (i.e., *“Have you ever been involved in a crash while riding an e-scooter?”* 0 = no crash, and 1 = any crash), and associated injuries (i.e., *“Were you injured in the crash?”* 0 = non-injury crash, and 1 = injury crash). In this study, an e-scooter-related crash was defined as *“impacting any vehicle type, a pedestrian, bike, another scooter, or any other incident due to riders losing control of an e-scooter (e.g., a crash on rough terrain or with fixed objects, etc.)”*. Participants were also asked to report *“any sustained injuries at or above the severity of an abrasion, bruise, or sprain that were associated with an e-scooter-related crash”*. Additional queried injury characteristics included the type of injury, any required medical attention, and police record filing.

Among all exposure variables, the infrastructure selection was of particular interest. Participants were asked *“What percentage of your typical e-scooter trips are ridden on [relevant infrastructure]…”* Seven types of these infrastructures were queried, including a sidewalk, a protected bike lane (e.g., a physical barrier existed between the rider and vehicle traffic), an unprotected bike lane, and shoulder or vehicle lane of a neighborhood or major street (e.g., higher potential for motor-vehicle exposure). For analysis purposes, reduced categories of relevant exposures of interest were included in the final statistical models, including using binary categories for each infrastructure exposure (i.e., the threshold of riding frequency is set to be greater than 25% of time).

Two questions measured helmet use, namely, *“The last time you rode an e-scooter, did you wear a helmet?”*, and *“How frequently do you wear a helmet when you ride an e-scooter?”* At the crash level, relevant factors were also obtained, including season, time of day, weather condition, crash location (e.g., infrastructure), perceived causes of crashes, and occupancy of the hands at the time of crash. Furthermore, participants were also surveyed about their safety preferences (e.g., *“Where do you most prefer to ride an e-scooter?”*), and avoidance of any road infrastructure when riding an e-scooter (e.g., *“Do you change your route or try to avoid any of the following types of infrastructure when riding an e-scooter?”*).

### 2.4. Statistical Analysis

Descriptive statistics were provided to identify the frequencies and percentages of those who reported having been involved in an e-scooter-related crash (i.e., any crash), injury crash, and no crash, within each exposure subcategory. The associations between various person-level, behavioral, and crash-level factors and the risks of the rider’s involvement in an e-scooter-related crash and injury crash were estimated by calculating the risk ratios (RRs) and their corresponding 95% confidence intervals (CIs) using the Negative Binomial (NB) regression models. Poisson robust variance regressions were applied when there was a convergence issue using the NB method [35,36]. For each exposure of interest, relevant confounding variables were controlled for as covariates in the models. Because few participants (2.8%) reported more than one crash, the analysis only included the first reported incident. Further descriptive analyses were performed for injury characteristics and riders’ safety perceptions of different infrastructures. All statistical analyses were conducted with SAS version 9.4 (SAS Institute, Cary, NC, USA).

## 3. Results

### 3.1. Descriptive Statistics

A total of 58 riders (27.6% of 210) reported having been involved in an e-scooter-related crash (any type), with 32 crashes (15.2%) resulting in an injury outcome. The most prevalent rider characteristics were males (70%), in the age range of 26–40 years old (47.1%), lived in the U.S. (61.0%), and reported having ever ridden an e-scooter a total of 21+ times (60%) or typically rode an e-scooter on a daily or almost daily basis (50.5%). As shown in Table 1, 86.2% of females had not experienced an e-scooter-related crash, compared to 67.3% of males. Among the most frequent users of e-scooters, approximately two-fifths reported having been involved in an e-scooter-related crash, and one in five riders had experienced an e-scooter-related injury (See Table 1).

Riders’ typical e-scooter trips were more frequently ridden in a protected bike lane (41.1% reported greater than 25% of the time), followed by the sidewalk (39.0% reported), and then the shoulder of a neighborhood street or an unprotected bike lane (34.8% reported) (see Table 1). Less commonly selected infrastructures were the shoulder (13.5% reported greater than 25% of time) and vehicle lanes (10.6% reported) of the major street (Table 1). More than half of the riders indicated helmet use for the entire ride of their most recent e-scooter trip or used a helmet frequently or always while riding an e-scooter—about 40% of them had an e-scooter-related crash, and 20% had an injury crash (See Table 1).

### 3.2. Descriptive Statistics

Table 2 summarizes the associations between riders’ personal and behavioral factors and their self-reported involvements in an e-scooter-related crash or injury crash. Females had a significantly lower risk of being involved in an e-scooter-related crash compared to males (RR = 0.44, 95% CI = 0.22 to 0.89); however, they were more prone to an injury outcome from these crashes (RR = 1.52, 95% CI = 1.02 to 2.26). A significantly elevated e-scooter-related crash risk was observed as riders’ total usage of e-scooters or riding frequency increased (RR = 4.25 or RR = 3.01, respectively, in Table 2). Neither of the riders’ age range or country was found to be significantly associated with their e-scooter-related crash or injury risk (See Table 2).

Specifically, the risk of being involved in an e-scooter-related injury crash was almost twice as great among riders whose typical e-scooter trips were ridden on the sidewalk for more than 25% of the time than those who did not (RR = 2.05, 95% = 1.02 to 4.16). In contrast, riding more regularly on a protected or unprotected bike lane demonstrated a potentially significant protective effect on reducing riders’ risk of being involved in an e-scooter-associated injury crash (RR = 0.41 and 0.50, respectively). Additionally, participants who reported having used a helmet for their most recent e-scooter ride or wearing a helmet more frequently while riding an e-scooter had a greater risk for an e-scooter-related crash of any type than those who did not (See Table 2).

### 3.3. Crash Related Factors and Injury Characteristics

At the crash level, an e-scooter-related crash was more likely to be associated with an injury outcome when it occurred on non-paved surfaces (i.e., parking lot, gravel road, unpaved bike trials in the park, etc.), compared to the bike lane (i.e., either protected or unprotected) (RR = 2.66, 95% CI = 1.35 to 5.27). Although not statistically significant, it is worth noting that the risk of an e-scooter-related injury was nearly twice as great for crashes that occurred on the sidewalk and the shoulder or vehicle lane of a neighborhood street, in reference to the bike lane (see Table 3 for the borderline significant results). In addition, an e-scooter-related injury crash was more likely when the crash occurred during summer, in the afternoon, under clear weather conditions, and as a result of hazardous road surfaces, loss of balance, or scooter malfunctions (See Table 3). However, none of these associations were statistically significant in the present analyses.

As shown in Table 4, 75.0% of e-scooter-related injuries in this study were characterized as soft tissue injury (i.e., scrape, cut, bruise), followed by 18.8% being orthopedic injury (i.e., broken bone). Approximately 62.5% of the participants reported they had never sought medical attention to treat those e-scooter-related injuries—only one injury crash case was filed by the police (See Table 4).

### 3.4. Riders’ Preferences, Perceived Safety, and Avoidance of Road Infrastructures

Moreover, in Table 5, the protected bike lane was perceived as the most preferred road infrastructure for riding an e-scooter by 51.9% of the riders, followed by the sidewalk (14.3%). Similarly, approximately 66.7% perceived the protected bike lane, and 17.6% perceived the sidewalk as the safest infrastructure for e-scooter trips (Table 5). The least favorable types of infrastructures where a rider would try to avoid or change the route included the major street with lots of traffic and activity (71.0%), followed by streets with hazardous surfaces (e.g., potholes) (69.1%), and streets with gravel roadway (55.7%) (Table 5).

## 4. Discussion

This study investigated risk factors associated with self-reported involvement in e-scooter-related crashes and injury crashes among e-scooter riders. Despite a small sample size, a relatively high prevalence of e-scooter-related crashes or injuries were reported (i.e., 26.7% and 15.2%, respectively). Due to the intrinsically different sources of populations and communities, it is not practical to directly compare the magnitude of this risk to other e-scooter-related injury case studies heavily relying on medical records such as trauma registries, emergency department encounters, other outpatient or clinic data, etc. [12,13,14,15,16,17]. Given the scarce e-scooter-related crash statistics and safety research, this finding may be indirectly supported by similar studies investigating self-reported crashes and injuries involving e-bikes or bicycles (i.e., in single-unit crashes) among relevant user groups [37,38].

In this study, gender differences were presented in crash and injury risk with e-scooters. The study showed males had a greater risk of being involved in an e-scooter-related crash, likely because they were in general more frequent riders, had fewer perceived concerns about e-scooter travel safety, and were more prone to risk-taking behaviors while riding an e-scooter [3,28,38,39]. Consistently, Yang and colleagues suggested males were overrepresented in 167 U.S. e-scooter incidents through the mining of news reports from 2017 to 2019 [32]. Furthermore, the present study revealed females had a 1.5 times higher risk of being involved in an e-scooter-related injury crash than males. This finding was contrary to previous studies where male patients were shown to disproportionally account for e-scooter-related injuries, particularly in crashes involving motor vehicles [13,17,21,22]. This finding may suggest a need to examine potential e-scooter design issues that may have better male anthropometric accommodations but are maladaptive to female riders in aspects such as handlebar height, center of gravity, and required upper body strength. However, it should be noted that the majority of self-reported crashes in the present study were subject to the more frequently occurring and minor injury types that may not necessarily require medical attention. Other reasons such as reporting bias, lack of experience (e.g., more injuries resulting from novice use), or sometimes being less adept at riding an e-scooter among female riders may also help explain the gender difference in the e-scooter-related injury risk [31,32].

Riders’ age or country was not found to significantly affect their self-reported e-scooter-related crash and injury involvement. Using the U.S. National Electric Injury Surveillance System (NEISS) data, a recent study identified that approximately 60% of motor-vehicle-involved e-scooter-related injuries were attributable to those aged 18–39 years old, followed by 40–64 years old (25.3%) [17]. In contrast, approximately 80% of e-scooter-related injuries that did not involve a motor vehicle occurred for those less than 40 years old (i.e., 41.2% for the age range of 18–39 and 35.4% for the age range of <18) [17]. Future attention to e-scooter safety may be directed to those less-advantaged communities or groups (including older riders who were typical more fragile and vulnerable), where there is a lack of access to affordable mobility, and who have greater interests in substituting public transportation with shared mobility options [2,6,19]. Primarily, the general accessibility of infrastructures such as protected bike lanes and the varying state of e-scooter legislation development could have led to potential differences in e-scooter-related crash or injury risks between the U.S. and other countries, but this expected result was not observed in this study [8,24,25]. However, accessibility issues may have, in part, contributed to the different rates of the more severe hospital e-scooter-related injuries as presented in other studies [11,13,15,22].

Notably, the study demonstrated that more frequent e-scooter trips ridden on bike lanes, either protected or unprotected, were a protective factor for reducing e-scooter-related injuries. In line with findings from other survey research (e.g., New Jersey, Austin), protected bike lanes received the highest preference among e-scooter riders and, in turn, were also perceived as the safest road infrastructure for e-scooter trips [10,31]. In contrast, an increased frequency (i.e., greater than 25% of the time) of e-scooter riding on the sidewalks was associated with nearly double the risk of self-reported e-scooter-related injury crashes. Many other e-scooter studies also provided support for this finding [33,34,40]. For instance, Badeau and colleagues reported sidewalks accounted for the majority of e-scooter-related injury types (i.e., 44% and primarily minor injuries) [40]. Another study suggested that e-scooters riders were more likely to encounter multiple times greater numbers of vibration events per mile ridden when riding on concrete sidewalks in the neighborhood than when riding a bike or on asphalt pavement conditions [33]. Using emergency department admission data in Washington, DC, a previous study reported that 58% of 105 e-scooter injuries occurred on the sidewalk, compared to 23% on the road [34].

The commonly documented sources of e-scooter-related injuries on sidewalks is likely to arise from single crashes (i.e., falls) on hazardous surfaces, such as pebbles, potholes, or cracks, and when transitioning from sidewalks to other road infrastructures [20,29,31,34,41]. In addition, riding e-scooters on sidewalks could also create potential hazards to pedestrians, such as blocking pedestrian right-of-way, abrupt overtaking at high speeds, and frequently underreporting conflicts [9,42,43]. Although more and more legislations have prohibited e-scooters from sidewalks, interestingly, many e-scooter riders were still observed to consistently and repeatedly ride on sidewalks [8,18,20]. Likewise, a substantial proportion of riders in this study also perceived sidewalks as the safest and most preferred infrastructure for riding an e-scooter. These controversial observations can be due to a combination of factors, including fear of encountering motor vehicles (e.g., as evidenced in this study, the avoidance of major streets with lots of activity and traffic), inaccessibility to bike lanes, and possibly low rider awareness of e-scooter laws [20,43]. In fact, e-scooter-related injuries involving motor vehicles were primarily found to occur on the street and were typically more severe than injuries from motor-vehicle-uninvolved crashes [17,43]. Further research is needed to investigate how bike lanes and other types of segregated transitways, if properly installed, would influence the interactions between e-scooters and a mix of traffic (e.g., bikes, other e-scooters, or potentially motor vehicles) under various riding circumstances (e.g., intersection maneuvers, also see [44]).

Importantly, this study provided evidence for an elevated risk of e-scooter-related crashes occurring on sidewalks, in the neighborhood street, and on non-paved surfaces, compared to bike lanes. This result further supported the riders’ infrastructure selection findings mentioned previously. The reported causes of e-scooter crashes also mirrored what has been documented in other studies, such as loss of balance or scooter malfunction and hazardous road surface conditions (which is also a primary reason for riders’ perceived avoidance of any infrastructure or change route in this study) [20,29,31,41]. Like the directionality of results in this study, some evidence also suggested that e-scooter-related crashes were more likely to occur during summer [15,32], or with inadequate lighting [44]. However, it is still unclear whether evening hours and certain weather conditions, such as precipitation or snow/ice, would increase the risk of e-scooter-related crashes or injuries. Surprisingly, helmet use was associated with more e-scooter-related crashes in this study. However, it should be noted that more than half of participants in this study reported having been wearing a helmet, compared to nil or relatively uncommon helmet use among e-scooter users from other studies, particularly for shared e-scooters [20,21,45]. Due to the lack of helmet use exposure on each infrastructure type and trip-specific information, it is challenging to reliably draw a clear relationship between riders’ helmet use frequencies and the outcomes in the present study. There also exists a research gap in establishing large-scale e-scooter-related crash-reporting databases that would identify critical crash attributes, including rider behavioral and road environment factors that can be potentially modified to develop preventive measures and guide policy making.

### Strengths and Limitations

The major scientific contribution of this study was to identify the characteristics of and ascertain risk factors for electric scooter-related crashes and injury crashes, which to the researchers’ best knowledge, was less investigated in prior literature. The results provided important practical implications for enhancing micromobility across multiple stakeholders. For instance, the results present an urgent call for transportation engineers and local road planners to take preventive action and develop cost-effective solutions for achieving a robust network of interconnected bike lanes in the urban areas. Another important contribution of this study was that it revealed a mismatch between a high preference for sidewalks perceived by e-scooters and the actual risk for e-scooter-related injuries associated with riding on sidewalks. This finding further demonstrates a demanding need for legislators, policymakers, or local road safety agencies to further assess the reasons why riders prefer the sidewalk, and thus plan for appropriate behavioral or legislative intervention strategies to promote safer communities for e-scooter riding. Moreover, the findings may suggest opportunities for design improvements to e-scooters to better handle poor surface conditions, reducing one-handed riding (i.e., support signaling and drink or bag carrying), and, most importantly, examine anthropometric differences in riders (i.e., gender and age) to ensure safe riding and injury prevention.

There are several limitations to this study. First, the study was subject to a small sample size and a limited number of crash and injury observations (including rare crashes involving a motor vehicle), potentially restricting the generalizability of results to other rider populations. Secondly, the shared or unshared status of e-scooters was unknown, which could have modified the interpretations of the study findings within each user group [2,6,7,9,28]. Potential sources of biases in the results could be due to the limitations of the cross-sectional study design for causal inferences, participants’ recall biases (particularly regarding the categorization of riding frequencies on each infrastructure), missing responses, and other factors [46,47]. The “survivor effect” may also be represented, in which those who suffered from the most severe or fatal e-scooter-related injuries were less likely to be sampled in this study [48]. Other unmeasured confounding variables, such as riders’ skills, scooter equipment (e.g., size, speed, brakes), perceived risks, and disparities in legislation status, could also have affected riders’ self-reported risk of e-scooter-related crashes and injury crashes [8,9,18,24,25,42]. Finally, although not strictly a limitation because this study focused on a non-hospitalized population, e-scooter-related injuries in this study could differ from other hospital-data-based studies in terms of many aspects, including the injury mechanism, injured body parts, and other characteristics, because only minimal to moderate levels of injury severities were captured.

## 5. Conclusions

This study analyzed various risk factors associated with self-reported involvements in e-scooter-related crashes and injury crashes among a cohort of e-scooter riders. The exposures of interest consisted of rider demographics, riding behaviors pertinent to riders’ infrastructure selection, helmet use, and other crash-relevant characteristics. Specifically, an elevated risk of e-scooter-related crashes (any type) was found among male riders and frequent users of e-scooters. Moreover, the primary protective factor for e-scooter-related injury crashes was riding on bike lanes, either protected or unprotected. In contrast, relevant risk factors included female riders, who reported more frequently riding on sidewalks, and when a crash occurred on non-paved surfaces.

The findings may help inform public policy regarding e-scooter legislation and prioritize efforts to establish suitable road infrastructures, in particular bike lanes, for improved e-scooter riding safety. Specifically, future legislation could enhance public education (e.g., advocating the use of protective equipment), require additional training for first-time riders, and develop clear sanctions for e-scooter traffic violations (e.g., riding on sidewalks). Establishing proactive, informed legislation that accounts for the safety needs of e-scooter riders, rather than reactive, trial-and-error legislative approaches, can help to prevent conflicts surrounding e-scooter riders [49]. In addition, further injury surveillance research is also needed to systematically track e-scooter-related injuries among other vulnerable road user populations, such as pedestrians, cyclists, or other e-scooter riders.

## Figures and Tables

**Figure 1 ijerph-19-10129-f001:**
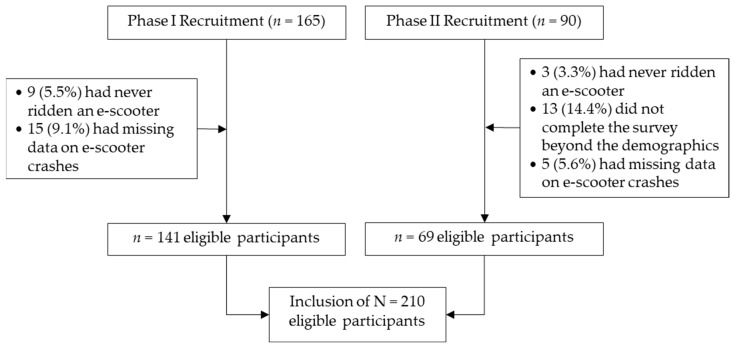
A flowchart of participant recruitments and eligibility.

**Table 1 ijerph-19-10129-t001:** Scooter rider demographics, riding frequency and locations, and helmet use among those who reported having been involved or not involved in an e-scooter related crash.

	Total	No Crash		Any Crash		Injury Crash	
	*N*	*n*	%	*n*	%	*n*	%
**Rider demographics**							
***Age***							
18–25	56	42	*75.0*	14	*25.0*	9	*16.1*
26–40	99	72	*72.7*	27	*27.3*	11	*11.1*
41–64	52	37	*71.2*	15	*28.8*	11	*21.2*
65+	3	1	*33.3*	2	*66.7*	1	*33.3*
***Gender***							
Male	147	99	*67.3*	48	*32.7*	24	*16.3*
Female	58	50	*86.2*	8	*13.8*	7	*12.1*
Non-binary	5	3	*60.0*	2	*40.0*	1	*20.0*
***Location***							
United States	128	97	*75.8*	31	*24.2*	16	*12.5*
Other countries	82	55	*67.1*	27	*32.9*	16	*19.5*
**E-scooter riding frequency**							
***Total times of ever riding an e-scooter***							
1–5	50	48	*96.0*	2	*4.0*	2	*4.0*
6–10	20	16	*80.0*	4	*20.0*	4	*20*
11–20	13	12	*92.3*	1	*7.7*	0	*0.0*
21+	127	76	*59.8*	51	*40.2*	26	*20.5*
***Frequency of riding an e-scooter***							
Daily or almost daily	106	62	*58.5*	44	*41.5*	24	*22.6*
Weekly	36	27	*75.0*	9	*25.0*	4	*11.1*
Monthly	12	11	*91.7*	1	*8.3*	1	*8.3*
Less than monthly	56	52	*92.9*	4	*7.1*	3	*5.4*
**Infrastructure type ^a^**							
***On the sidewalk***							
<1% of the time	41	28	*68.3*	13	*31.7*	6	*14.6*
1–25% of the time	45	35	*77.8*	10	*22.2*	5	*11.1*
26–50% of the time	28	25	*89.3*	3	*10.7*	3	*10.7*
51–75% of the time	11	11	*100.0*	0	*0.0*	0	*0.0*
75–99% of the time	12	8	*66.7*	4	*33.3*	3	*25.0*
100% of the time	4	3	*75.0*	1	*25.0*	1	*25.0*
***In a protected bike lane***							
<1% of the time	49	41	*83.7*	8	*16.3*	6	*12.2*
1–25% of the time	34	26	*76.5*	8	*23.5*	5	*14.7*
26–50% of the time	35	27	*77.1*	8	*22.9*	5	*14.3*
51–75% of the time	16	10	*62.5*	6	*37.5*	2	*12.5*
75–99% of the time	7	6	*85.7*	1	*14.3*	0	*0.0*
100% of the time	0	0	*0.0*	0	*0.0*	0	*0.0*
***In an unprotected bike lane***							
<1% of the time	35	27	*77.1*	8	*22.9*	5	*14.3*
1–25% of the time	57	46	*80.7*	11	*19.3*	8	*14.0*
26–50% of the time	25	19	*76.0*	6	*24.0*	3	*12.0*
51–75% of the time	20	16	*80.0*	4	*20.0*	1	*5.0*
75–99% of the time	2	1	*50.0*	1	*50.0*	0	*0.0*
100% of the time	2	1	*50.0*	1	*50.0*	1	*50.0*
***On the shoulder of a major street***							
<1% of the time	56	49	*87.5*	7	*12.5*	5	*8.9*
1–25% of the time	55	42	*76.4*	13	*23.6*	7	*12.7*
26–50% of the time	14	10	*71.4*	4	*28.6*	2	*14.3*
51–75% of the time	10	6	*60.0*	4	*40.0*	3	*30.0*
75–99% of the time	4	2	*50.0*	2	*50.0*	1	*25.0*
100% of the time	2	1	*50.0*	1	*50.0*	0	*0.0*
***On the shoulder of a neighborhood street***							
<1% of the time	24	18	*75.0*	6	*25.0*	5	*20.8*
1–25% of the time	68	56	*82.4*	12	*17.6*	5	*7.4*
26–50% of the time	26	17	*65.4*	9	*34.6*	6	*23.1*
51–75% of the time	15	13	*86.7*	2	*13.3*	2	*13.3*
75–99% of the time	5	4	*80.0*	1	*20.0*	0	*0.0*
100% of the time	3	2	*66.7*	1	*33.3*	0	*0.0*
***In the vehicle lane of a major street***							
<1% of the time	79	67	*84.8*	12	*15.2*	9	*11.4*
1–25% of the time	42	28	*66.7*	14	*33.3*	6	*14.3*
26–50% of the time	5	5	*100.0*	0	*0.0*	0	*0.0*
51–75% of the time	8	7	*87.5*	1	*12.5*	1	*12.5*
75–99% of the time	5	2	*40.0*	3	*60.0*	2	*40.0*
100% of the time	2	1	*50.0*	1	*50.0*	0	*0.0*
***In the vehicle lane of a neighborhood street***							
<1% of the time	30	27	*90.0*	3	*10.0*	2	*6.7*
1–25% of the time	69	52	*75.4*	17	*24.6*	10	*14.5*
26–50% of the time	18	11	*61.1*	7	*38.9*	5	*27.8*
51–75% of the time	14	13	*92.9*	1	*7.1*	0	*0.0*
76–99% of the time	6	4	*66.7*	2	*33.3*	1	*16.7*
100% of the time	4	3	*75.0*	1	*25.0*	0	*0.0*
**Helmet use**							
***Helmet use for the most recent ride***							
Yes, for the entire ride	115	71	*61.7*	44	*38.3*	23	*20.0*
Yes, for part of the ride	0	0	*0*	-	*-*	-	*-*
No	95	81	*85.3*	14	*14.7*	9	*9.5*
***Frequency of helmet use***							
Never	58	50	*86.2*	8	*13.8*	6	*10.3*
Very Rarely	9	8	*88.9*	1	*11.1*	0	*0.0*
Rarely	8	7	*87.5*	1	*12.5*	0	*0.0*
Occasionally	17	15	*88.2*	2	*11.8*	1	*5.9*
Very frequently	26	15	*57.7*	11	*42.3*	5	*19.2*
Always	92	57	*62.0*	35	*38.0*	20	*21.7*

^a^ Data indicated self-reported percentage of time a rider’s typical e-scooter trips were ridden on relevant infrastructure. Data were only available for the first phase of recruitment.

**Table 2 ijerph-19-10129-t002:** Associations between scooter rider demographics, riding frequency and locations, and helmet use and the riders’ self-reported involvement in an e-scooter related crash.

	Any vs. No Crash		Injury vs. Non-Injury Crash	
	Adjusted	95% CIs	Adjusted	95% CIs
	RR ^a^		RR ^b^	
**Rider demographics**				
***Age range***				
18–25	1.0	--	1.0	--
26–40	1.06	0.60, 1.86	0.73	0.40, 1.32
41–65+	1.09	0.59, 2.03	1.02	0.60, 1.74
***Gender***				
Male	1.0	--	1.0	--
Female	**0.44**	**0.22, 0.89**	**1.52**	**1.02, 2.26**
***Country***				
United States	1.0	--	1.0	--
Other countries	1.25	0.80, 1.95	1.11	0.71, 1.73
**Riding frequency**				
***Total times of ever riding an e-scooter***				
1–20	1.0	--	1.0	--
21+	**4.25**	**1.92, 9.43**	0.89	0.48, 1.63
***Frequency of riding an e-scooter***				
Less frequent than daily	1.0	--	1.0	--
Daily or almost daily	**3.01**	**1.63, 5.58**	1.49	0.79, 2.82
**Infrastructure Type**				
***On the sidewalk***				
1–25% of the time	1.0	--	1.0	--
26%+ of the time	0.66	0.29, 1.47	**2.05**	**1.02, 4.16**
***In a protected bike lane***				
1–25% of the time	1.0	--	1.0	--
26%+ of the time	1.19	0.63, 2.25	**0.41**	**0.21, 0.78**
***In an unprotected bike lane***				
1–25% of the time	1.0	--	1.0	--
26%+ of the time	0.99	0.54, 1.83	**0.50**	**0.26, 0.96**
***On the shoulder of a major street***				
1–25% of the time	1.0	--	1.0	--
26%+ of the time	1.38	0.72, 2.65	0.95	0.42, 2.13
***On the shoulder of a neighborhood street***				
1–25% of the time	1.0	--	1.0	--
26%+ of the time	1.03	0.52, 2.03	0.93	0.47, 1.85
***In the vehicle lane of a major street***				
1–25% of the time	1.0	--	1.0	--
26%+ of the time	0.86	0.38, 1.95	1.17	0.51, 2.69
***In the vehicle lane of a neighborhood street***				
1–25% of the time	1.0	--	1.0	--
26%+ of the time	0.98	0.49, 1.97	0.93	0.54, 1.61
**Helmet use**				
***Helmet use for the most recent ride***				
Yes, for the entire ride	**2.04**	**1.12, 3.68**	1.05	0.59, 1.88
No	1.0	--	1.0	--
***Frequency of helmet use***				
Very frequently or Always	**2.41**	**1.24, 4.69**	1.15	0.55, 2.42
Not frequently	1.0	--	1.0	--

**RR: Risk Ratio.**^a^ Models adjusted for age range, gender, and country. ^b^ Models adjusted for age range, gender, country, and total riding frequency.

**Table 3 ijerph-19-10129-t003:** Associations between crash-level factors and e-scooter related injury and non-injury crashes.

	Number (n) of Injured among Any Crash (N)			Injury vs. Non-Injury Crash	
	N	n	%	Adjusted	95% CI
				RR ^a^	
**Crash characteristics**					
**Season of crashes**					
Summer	10	8	80.0	1.30	0.80, 2.12
Other seasons	46	24	52.2	1.0	--
**Time of day**					
6:00 a.m.–12:00 p.m.	19	9	47.4	1.0	--
12:00 p.m.–6:00 p.m.	21	15	71.4	1.57	0.83, 2.96
6:00 p.m.–6:00 a.m.	14	8	57.1	1.02	0.53, 1.94
**Weather condition**					
Clear (no notable weather conditions)	45	29	64.4	1.0	--
Other weather conditions	9	3	33.3	0.54	0.21, 1.37
**Location of crashes**					
Sidewalk	12	9	75.0	1.87	0.97, 3.60
Bike lane (protected or unprotected)	13	5	38.5	1.0	--
Shoulder or vehicle lane of a major street	9	3	33.3	0.89	0.30, 2.63
Shoulder or vehicle lane of a neighborhood street	12	9	75.0	2.01	0.93, 4.33
Other, includes non-paved surfaces. ^b^	8	6	75.0	**2.66**	**1.35, 5.27**
**Perceived causes of crashes**					
**Related to a moving motor vehicle**					
Yes	9	1	11.1	0.20	0.23, 1.18
No	45	31	68.9	1.0	--
**Related to a non-moving motor vehicle**					
Yes	3	1	33.3	0.47	0.12, 1.91
No	51	31	60.8	1.0	--
**Related to a none-motor vehicle (i.e., bicycle, e-scooter, pedestrian)**					
Yes	8	4	50.0	0.74	0.39, 1.40
No	46	28	60.9	1.0	--
**Loss of balance or scooter malfunction**					
Yes	17	12	70.6	1.20	0.73, 1.96
No	37	20	54.1	1.0	--
**Road surface conditions (i.e., gravel, icy, sandy, slippery surfaces, potholes, etc.)**					
Yes	24	17	70.8	1.36	0.86, 2.14
No	30	15	50.0	1.0	--
**Other reasons ^c^**					
Yes	6	4	66.7	1.12	0.61, 2.05
No	48	28	58.3	1.0	--
**Hands on the e-scooter’s handlebars**					
Yes	49	29	59.2	1.0	--
No (e.g., holding something or “signaling” with one hand)	5	3	60.0	0.84	0.40, 1.80

^a^ Models adjusted for age range, gender, country, and riding frequency. ^b^ Non-paved surfaces included parking lot, gravel road, unpaved bike trails in the park, and others. ^c^ Other perceived reasons included no training on riding, user error, turning too sharply, and did not recall.

**Table 4 ijerph-19-10129-t004:** Injury characteristics for e-scooter related injury crashes.

Total N of Injury Crashes = 32		
	n	% ^a^
**Injury type ^b^**		
Soft tissue injury (i.e., scrape, cut, bruise)	24	75.0
Orthopedic injury (i.e., broken bone)	6	18.8
Dental injury	2	6.3
Head injury (i.e., concussion)	1	3.1
Other, please describe ^c^	3	9.4
Did not recall	1	3.1
**Sought medical attention**		
No	20	62.5
Yes, I went to a primary care clinic or urgent care	5	15.6
Yes, I went to the emergency room	5	15.6
Other, please describe ^d^	1	3.1
Did not recall	1	3.1
**Police filing**		
No	31	96.9
Yes	1	3.1

^a^*p* < 0.0001 for all chi-squared tests for equal distributions. ^b^ The ns do not add up to the total N due to some participants reporting multiple injury types for one single crash incident. ^c^ Other injury types included knee injury (i.e., anterior cruciate ligament), helmet hit the handlebar of another e-scooter, and road rash. ^d^ Other medical attention included one participant seeing a traumatologist one week after the injury.

**Table 5 ijerph-19-10129-t005:** Riders’ perceptions of various types of infrastructures.

	Total N of Participants = 210	
	n	% ^a^
**Most preferred infrastructure when riding an e-scooter**		
In the vehicle lane of a major street	9	4.3
In the vehicle lane of a neighborhood street	17	8.1
On the shoulder of a major street	3	1.4
On the shoulder of a neighborhood street	23	11.0
Protected bike lane	109	51.9
Sidewalk	30	14.3
Unprotected bike lane	19	9.1
**Infrastructure perceived to be the safest when riding an e-scooter**		
In the vehicle lane of a major street	5	2.4
In the vehicle lane of a neighborhood street	12	5.7
On the shoulder of a major street	0	0
On the shoulder of a neighborhood street	10	4.8
Protected bike lane	140	66.7
Sidewalk	37	17.6
Unprotected bike lane	6	2.9
**Types of infrastructure avoided (multiple responses allowed)**		
I do not change my route to avoid any particular types of infrastructure.	23	11.0
Major street with lots of traffic and activity	149	71.0
Streets with gravel roadway	117	55.7
Streets with hazardous surfaces (e.g., potholes)	145	69.1
Non-signalized intersections	25	11.9
Streets without a sidewalk	51	24.3
Streets without a shoulder	66	31.4
Signalized intersections	22	10.5
Others, please describe.	12	5.7

^a^*p* < 0.0001 for all chi-squared tests for equal distributions.

## Data Availability

Data may be obtained upon reasonable request and are not publicly available.

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
