# Peer review of "Characteristics and Risk Factors for Electric Scooter-Related Crashes and Injury Crashes among Scooter Riders: A Two-Phase Survey Study"

_ijerph, 2022, doi:10.3390/ijerph191610129_

Round 1

Reviewer 1 Report

The risk factors associated with e-scooter-related crashes and injury crashes were investigated in this study. The paper is easy to follow. The analysis and results were appropriately presented. The only issue with this paper is the data validation section. The authors should provide additional details about how they obtained the data. Are the authors conducting naturalistic experiments or was the data collected using other methods? Instruments used to collect the data should also be described in detail.

Author Response

The authors greatly appreciate the commitment of the reviewers to provide a comprehensive review of this paper. Below, the reviewers’ comments are identified in Bold font and the authors’ responses are noted in Italics. All edits are red-tracked in the current revised manuscript.

Reviewer #1:

The risk factors associated with e-scooter-related crashes and injury crashes were investigated in this study. The paper is easy to follow. The analysis and results were appropriately presented. The only issue with this paper is the data validation section. The authors should provide additional details about how they obtained the data. Are the authors conducting naturalistic experiments or was the data collected using other methods? Instruments used to collect the data should also be described in detail.

                  Thank you for your comment! On Page 3, and in Section 2.2 Data Collection, the authors specified the methods for recruitment and data collection as “…Participants were recruited through posted announcements seeking research volunteers on multiple social media platforms and technology discussion forums specific to e-scooters or micromobility, such as various e-scooter Facebook groups, Reddit webpages, and the research laboratory Twitter, etc. Upon recruitment, participants consented to the study and were provided with a secured link to an online survey questionnaire via the Qualtrics software using the University of Minnesota design scheme (e.g., headings, back-ground, color, etc.)…”

                  To better illustrate the survey instrument used to collect the data, more details were provided based on your suggestion. Relevant references were also added regarding the development and validation of the survey. Please refer to the second paragraph, in Section 2.2, which was cited as “The survey questionnaire was developed based on a literature review of available e-scooter-related survey research [19,21,31], and investigations into the crash and injury characteristics involving e-scooters [12-17,32,33,34]. Participants were asked to respond to 116 single or multiple-choice questions regarding e-scooter riders' demographic characteristics, self-reported riding behaviors, helmet use, and involvement in e-scooter-related crashes and injury crashes (See Tables 1, 3, and 4 for more detailed question categories). The below section described definitions of variables in detail, as were measured in the survey questionnaire.” (See Page 3).  

                  Following this paragraph, the authors also modified relevant descriptions, throughout Section 2.3. Definitions of Variables. In the revised manuscript, the texts of important survey questions were provided in details using the quotation marks, and were also Italicized to help improve the clarity to the readers (Please refer to Section 2.3, on Page 3). 

Reviewer 2 Report

The research is devoted to the safety issues of such type of micromobility as e-scooter riding. The authors consider e-scooter riding with minor crash/injury cases with emphasis on gender aspect. The main attention is paid to the relation of path of riding – bike lanes, shoulders, and so on – and the possibility of cases of crash/injury.

The research is really urgent, since e-scooter riding is gaining more and more popularity all over the world, and correspondently the cases of crashes/injuries are growing, and it is very important to consider all possible factors impacting the safety of riding.

I suppose, some minor improvements can be done.

1.     The contribution, novelty, importance of the article to the scientific and/or practical area could be added.

2.     In Conclusions the authors say only few words about stakeholders – “The findings may help inform public policy…”. I suppose, the stakeholders and their specific interests may and should be listed.

3.     The “distribution” of references is not even; most of the references are in the introduction section; however, the discussion and the methodology sections do not provide enough references to other scholars’ researches.

Author Response

The authors greatly appreciate the commitment of the reviewers to provide a comprehensive review of this paper. Below, the reviewers’ comments are identified in Bold font and the authors’ responses are noted in Italics. All edits are red-tracked in the current revised manuscript.

Reviewer #2:

The research is devoted to the safety issues of such type of micromobility as e-scooter riding. The authors consider e-scooter riding with minor crash/injury cases with emphasis on gender aspect. The main attention is paid to the relation of path of riding – bike lanes, shoulders, and so on – and the possibility of cases of crash/injury.

The research is really urgent, since e-scooter riding is gaining more and more popularity all over the world, and correspondently the cases of crashes/injuries are growing, and it is very important to consider all possible factors impacting the safety of riding.

I suppose, some minor improvements can be done.

  1. The contribution, novelty, importance of the article to the scientific and/or practical area could be added.

                  The authors appreciate your feedback. Please refer to the revised first paragraph of Section 4.1 Strengths and Limitations for specific contribution, novelty and importance of the article, on Page 14.

                  The revised paragraph now includes “The major scientific contribution of this study was to identify the characteristics of and ascertain risk factors for electric scooter-related crashes and injury crashes, which to the researchers’ best knowledge, was less investigated in prior literature. The results provided important practical implications for enhancing micromobility across multiple stakeholders. For instance, there presents an urgent call for transportation engineers and local road planners to take preventive action and develop cost-effective solutions for achieving a robust network of interconnected bike lanes in the urban areas. Another important contribution for this study was it also revealed a mismatch between a high preference for sidewalks perceived by e-scooters and the actual risk for e-scooter-related injuries associated with riding on sidewalks. This finding further demonstrated a demanding need for legislators, policymakers, or local road safety agencies to further assess the reasons why riders prefer the sidewalk and thus plan for appropriate behavioral or legislative intervention strategies to promote safer communities for e-scooter riding. Moreover, the findings may suggest opportunities for design improvements to e-scooters to better handle poor surface conditions, reducing one-handed riding (i.e., support signaling and drink or bag carrying), and most importantly examine anthropometric differences in riders (i.e., gender and age) to ensure safe riding and injury prevention.” (See Page 14).

  1. In Conclusions the authors say only few words about stakeholders – “The findings may help inform public policy…”. I suppose, the stakeholders and their specific interests may and should be listed.

                  Thank you for your helpful insight! This was addressed in the second paragraph of the Conclusion section, which was quoted as “The findings may help inform public policy regarding e-scooter legislation and prioritize efforts to establish suitable road infrastructures, in particular bike lanes, for improved e-scooter riding safety. Specifically, future legislation could enhance public education (e.g., advocating the use of protective equipment), require additional training for first-time riders, and develop clear sanctions for e-scooter traffic violations (e.g., riding on sidewalks). Establishing proactive, informed legislation that accounts for the safety needs of e-scooter riders, rather than reactive, trial and error legislative approaches, can help to prevent conflicts surrounding e-scooter riders [49]. In addition, further injury surveillance research is also needed to systematically track e-scooter-related injuries among other vulnerable road user populations such as pedestrians, cyclists, or other e-scooter rider.” (See Page 15).

                  Additional edits that provides more details on the stakeholder’s interests were also provided in Section 4.2 Strengths and Limitation (See Page 14), as were noted in above question.

  1. The “distribution” of references is not even; most of the references are in the introduction section; however, the discussion and the methodology sections do not provide enough references to other scholars’ researches.

            As suggested by Reviewer #2, relevant references were added to the methodology and discussion sections to improve the balance of reference distributions. The reference list was modified and reordered, as appropriate.